# Validity study using factor analyses on the Defining Issues Test-2 in undergraduate populations

**Youn-Jeng Choi**[1]*, **Hyemin Han**[1], **Meghan Bankhead**[2], **Stephen J. Thoma**[1]

1 Department of Educational Studies in Psychology, Research Methodology and Counseling, University of Alabama, Tuscaloosa, Alabama, United States of America, 2 Department of Psychological Science, Kennesaw State University, Kennesaw, Georgia, United States of America

☯ These authors contributed equally to this work.
* ychoi26@ua.edu

**Data Availability Statement:** All relevant data are within the paper and its Supporting Information files.

**Funding:** The authors received no specific funding for this work.

## Abstract

### Introduction

The Defining Issues Test (DIT) aimed to measure one's moral judgment development in terms of moral reasoning. The Neo-Kohlbergian approach, which is an elaboration of Kohlbergian theory, focuses on the continuous development of postconventional moral reasoning, which constitutes the theoretical basis of the DIT. However, very few studies have directly tested the internal structure of the DIT, which would indicate its construct validity.

### Objectives

Using the DIT-2, a later revision of the DIT, we examined whether a bi-factor model or 3-factor CFA model showed a better model fit. The Neo-Kohlbergian theory of moral judgment development, which constitutes the theoretical basis for the DIT-2, proposes that moral judgment development occurs continuously and that it can be better explained with a soft-stage model. Given these assertions, we assumed that the bi-factor model, which considers the Schema-General Moral Judgment (SGMJ), might be more consistent with Neo-Kohlbergian theory.

### Methods

We analyzed a large dataset collected from undergraduate students. We performed confirmatory factor analysis (CFA) via weighted least squares. A 3-factor CFA based on the DIT-2 manual and a bi-factor model were compared for model fit. The three factors in the 3-factor CFA were labeled as moral development schemas in Neo-Kohlbergian theory (i.e., personal interests, maintaining norms, and postconventional schemas). The bi-factor model included the SGMJ in addition to the three factors.

### Results

In general, the bi-factor model showed a better model fit compared with the 3-factor CFA model although both models reported acceptable model fit indices.

**Competing interests:** The authors have declared that no competing interests exist.

## Conclusion

We found that the DIT-2 scale is a valid measure of the internal structure of moral reasoning development using both CFA and bi-factor models. In addition, we conclude that the soft-stage model, posited by the Neo-Kohlbergian approach to moral judgment development, can be better supported with the bi-factor model that was tested in the present study.

## Introduction

The Defining Issues Test (DIT) assesses how one defines the moral issues in a social problem [1–4]. Its theoretical framework is grounded on the theory of moral judgment development proposed by Kohlberg [5]. In the original version of the Kohlbergian model, the process of moral judgment development was explained in terms of three levels: the pre-conventional, conventional, and postconventional. One's level of moral judgment can be explained according to which moral philosophical rationale is employed while making moral judgments [6]. Those at the pre-conventional level are likely to make a moral decision based on the possibility of being punished or obtaining personal benefits. At the conventional level, a person tends to behave in a way that maintains a good relationship with others or social norms. Finally, a person who is situated at the postconventional level can critically evaluate the justifiability of existing laws and social norms based on universal moral principles. According to Kohlbergian theory, the most developed and sophisticated level of moral judgment is the postconventional level. Previous research has shown the presence of the higher level to be associated with moral motivation and behavior ([7]).

However, Neo-Kohlbergians, who revised the original Kohlbergian model, argued that although people generally tend to follow the developmental trajectory proposed by Kohlberg, they do not always rely on only one level of moral philosophical rationale while making moral judgments [1]. For instance, a person with highly developed moral reasoning does not always make moral judgments based on rationale at the post-conventional level. Instead, despite being more likely to use the post-conventional rationale, the person occasionally might refer to rationale from the pre-conventional or conventional level. As a way to cope with this limitation, the Neo-Kohlbergians proposed the soft-stage model, which differs from Kohlberg's original hard-stage model. According to this model, the development of moral judgment happens gradually, rather than like a discontinuous quantum jump between two stages, which was proposed in the original Kohlbergian theory [1]. The traditional Kohlbergian researchers were primarily interested in how transitions between levels (or stages) occur within persons, whereas Neo-Kohlbergians attempted to explain the process of moral judgment development in terms of how frequently the rationale based on the postconventional level is employed in moral decision-making across situations.

### The Defining Issues Test-2 (DIT-2)

The first DIT, the DIT-1, was developed based on the Neo-Kohlbergians' soft-stage model, and measures one's development of moral judgment in terms of the likelihood of the utilization of the postconventional schema [3]. Compared with the Moral Judgment Interview [8], which was developed based on the original Kohlbergian model and required extensive interviews for data collection, the DIT can be administered more easily because it is a paper-and-pencil test [1]. The DIT quantifies one's likelihood to utilize each individual schema (e.g., the

utilization of the postconventional level from 0 to 100%). It was designed to measure the likelihood of three schemas: personal-interest, maintaining-norms, and postconventional schemas. Given that it is feasible to implement and that it provides a quantified score of moral judgment development, the DIT has been widely used across different fields by researchers and educators to examine the trajectory of moral judgment development or to evaluate outcomes of moral educational programs (see [9] for the overview and https://ethicaldevelopment.ua.edu/about-the-dit.html for a list of studies applying the DIT).

Previous studies reported that the DIT measured three specific schemas of moral reasoning: the personal interests, maintaining norms, and postconventional schemas using universal moral principles [1,4,10–12]. Those who prefer the personal-interests schema are likely to focus on their personal stake and close personal relationships, and are likely to place less value on social norms and conventions. Those who prioritize the maintaining-norms schema tend to value existing social norms and conventions, thereby emphasizing a society-wide perspective. Those who tend to utilize the postconventional schema are likely to believe that "moral obligations are to be based on shared ideals, which are reciprocal and are open to debate and tests of logical consistency, and on the experience of the community" ([1], p. 307).

Rest, Cooper, Coder, Masanz, and Anderson [13] devised the DIT, and Rest, Narvaez, Thoma, and Bebeau revised it into the DIT-2 [2]. Unlike the DIT-1, which reports the score of the likelihood of the utilization of each individual schema, the outcome score of the DIT-2, the N2 score, demonstrates one's overall moral judgment development. The N2 score is calculated based on whether items corresponding to the postconventional schema are selected with higher priority than those corresponding to the personal-interests or maintaining-norms schemas [2]. Hence, in terms of consistency with the Neo-Kohlbergian theory, the DIT-2 would be more consistent than the DIT-1, given how the N2 score is calculated. The full measure and scoring information is available from the Center for the Study of Ethical Development. Information about the measure and scoring can be found at: https://ethicaldevelopment.ua.edu/about-the-dit.html.

The DIT has been tested and validated in various previous studies (see [14]). They have demonstrated that the DIT score possesses good predictive, convergent and divergent validity [9]. For instance, the DIT score is significantly changed by moral education and significantly predicts moral behavior (predictive validity), is significantly correlated with relevant cognitive and personality measures (convergent validity) and is significantly distinguishable from surveyed political orientations (divergent validity).

However, although the external validity of the DIT has been tested repeatedly, its internal validity, meaning whether or not its internal structure, which indicates construct validity, is well supported by data, has not been thoroughly tested in previous studies. Only a few prior studies have examined some aspects of the internal validity of the DIT. Several researchers conducting the previous validity studies [10,15,16] used exploratory factor analysis (EFA) with the principal component extraction method for the DIT-1; however, the DIT-2 has not been explored in this manner. A recent study evaluated the DIT-1 and DIT-2 with the item response theory [17]. Although van den Enden, Boom, Brugman, and Thoma reported results that supported the developmental structure of the DIT, they assumed that each level or each stage exists independent from the others. Another recent study tested the differential item functioning of the behavioral DIT [18], a revised version of the DIT that was designed for use in behavioral experiments [19]. As in the first study, the authors also assumed that the three schemas exist in the model as three independent factors. Although both these studies examined the multi-factor structure of the DIT, they did not perform confirmatory factor analysis (CFA) to directly evaluate its internal structure and validity.

## The present study

The purpose of this study was to validate the internal structure of DIT-2, which has not been well studied. Thus, we performed CFA based on the ordinary multi-factor model and bi-factor model to investigate the internal structure of DIT-2. Although the previous studies examined the internal structure of DIT-2 indirectly, they basically assumed that the three schemas were independent from each other as three factors; in other words, the previous analyses seemingly were based on the multi-factor model. Given that DIT-2 and the N2 score are conceptually grounded in the soft-stage model, which considers moral judgment development as a continuous process, testing the validity of DIT-2 with the multi-factor assumption, assuming that three schemas exist as three individual factors, would not be completely appropriate. Instead, a validity test should be based on the assumption that moral judgment development occurs continuously and a latent factor might well conceptually describe such a continuous development.

A bi-factor model assumes the existence of a general factor (G) that is explained by all items in addition to multiple individual latent factors in the structure of a measure [20]. The general factor assumes that all items in the measure explain one unidimensional factor as well as lower-level individual factors [21]. Given that the bi-factor model includes a unidimensional factor, it would be more consistent with the concept of the soft-stage model, which constitutes the conceptual basis of DIT-2. If three schemas exist in the form of a continuum as proposed by the soft-stage model, then it would be more plausible to examine the continuum as one unified dimension (as assumed in the bi-factor model) instead of three independent schemas. Although the bi-factor model seems similar to the higher-order model, which assumes that the relationships between items and the general factor are mediated by individual latent factors [20], we decided to employ the bi-factor model in our study because it is more conceptually coherent with the soft-stage model. The soft-stage model assumes that moral judgment development occurs continuously; thus, it would be reasonable to assume that all individual items have direct relations with the general factor if the factor is supposed to explain such a continuous developmental trend.

Next, we intended to compare model fit indicators between the bi-factor and ordinary multi-factor models to examine whether the bi-factor model is empirically better than the multi-factor model and if the soft-stage model can be better supported empirically. Given that the hypothetical general factor in our bi-factor model used to examine the DIT-2 was assumed to be the continuous development of moral judgment across different schemas (or levels), we used the term, Schema-General Moral Judgment (SGMJ), to explain the nature of the general factor in our study.

## Method

### DIT-2 scale

Five different moral dilemmas, or stories, about social problems were presented to the students, who were then asked to answer 12 five-point Likert-type questions (1 = no, 2 = little, 3 = some, 4 = much, 5 = great) and four ranking questions per story in order to measure what was important to them in decision-making about the social problem. The five dilemmas were labeled "Famine," "Reporter," "School Board," "Cancer," and "Demonstration." Nine of the total of 60 Likert-type questions were used to detect unreliable participants. Although ranking responses were also used to create a total score, only 51 Likert-type questions were used for factor analysis. An example question is "Isn't the doctor obligated by the same laws as everybody else if giving an overdose would be the same as killing her?" Three specific schemas of moral reasoning (the personal interests (PI) schema, the maintaining norms (MN) schema and the

postconventional (P) schema) were measured by patterns of ratings and rankings. Because the DIT-2 is protected by copyright, requests for access to the full scale should be addressed to the Center for the Study of Ethical Development (https://ethicaldevelopment.ua.edu/about-the-dit.html).

## Sample

We used the DIT-2 datasets provided by a third-party research center, the Center for the Study of Ethical Development at the University of Alabama. The samples were collected from undergraduate students in the United States between 2000 and 2009 (N = 44,742). To prevent culture and language bias, only students who were US citizens and used English as their primary language were selected. Ages ranged from 17 to 26; 47.2% of participants were male (n = 21,139) and 52% were female (n = 23,272). Regarding class year, 34.7% of the sample were seniors, 20.6% were juniors, 13.7% were sophomores, and 31.0% were freshmen. Total sample size for the factor analyses was 39,409 after the listwise deletion method was applied for handling missing values. Similar to the case of the full DIT-2 scale, the Center for the Study of Ethical Development retains the right to use the datasets, and requests for access to the datasets should be addressed to the center. The Center for the Study of Ethical Development is the repository of all raw DIT-2 data. These data are also available for use by researchers for secondary analyses. The DIT-2 datasets were collected by various researchers around the country, submitted for scoring to the Center for the Study of Ethical Development at The University of Alabama, and then added to the Center's database. Because of the secondary nature of the data, IRB approval was not obtained for the present study.

## Confirmatory factor analyses

Two different types of factor analyses were performed to evaluate the factor structures using SAS 9.4 statistical software: 3-factor CFA and bi-factor model (see Figs 1 and 2). Each item was assumed to measure one of three specific schemas of moral reasoning (i.e., PI, MN and P schemas). According to the DIT-2 Guide [22], the 51 rating items were assigned in the following manner, 20 PI items, 17 MN items, and 14 P items. PI-1 to PI-4, MN-1 to MN-2, and P-1 to P-3 were assigned to Famine. PI-5 to PI-8, MN-3 to MN-6, and P-4 to P-5 were assigned to

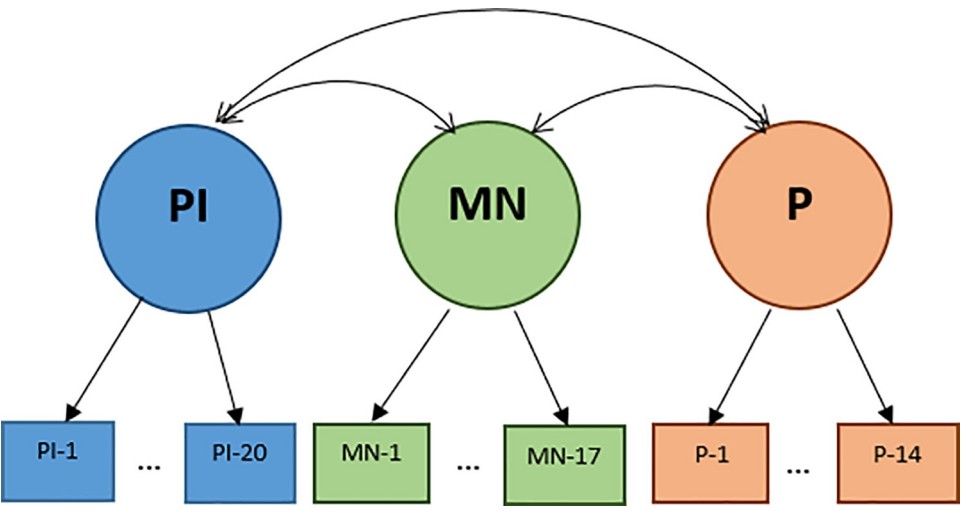

**Fig 1. Diagram for a 3-factor CFA model.**

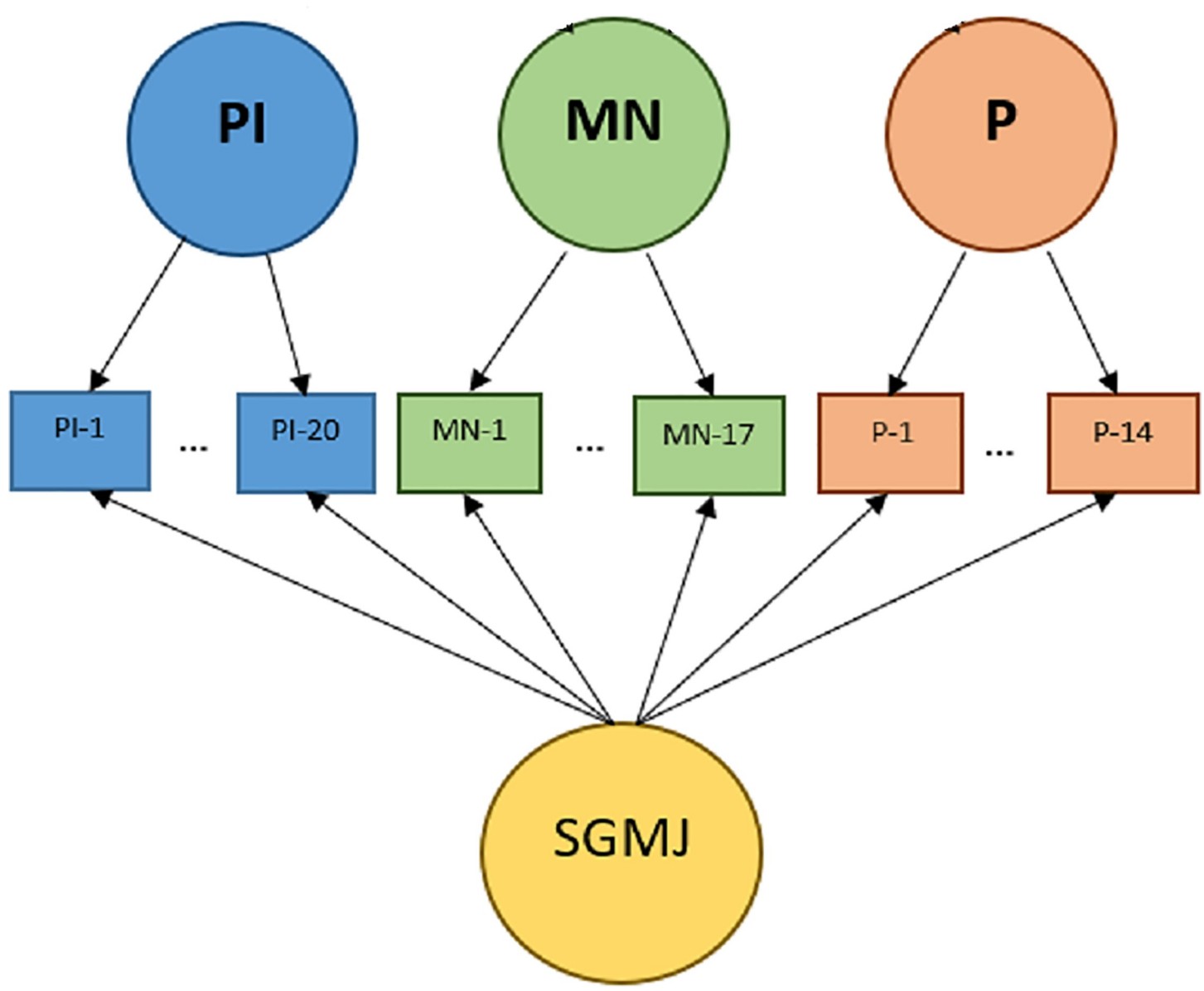

**Fig 2. Diagram for a Bi-factor model.**

Reporter. PI-9 to PI-12, MN-7 to MN-10, and P-6 to P-8 were assigned to School Board. PI-13 to PI-16, MN-11 to MN-13, and P-9 to P-11 were assigned to Cancer. PI-17 to PI-20, MN-14 to MN-17, and P-12 to P-14 were assigned to Demonstration.

A CFA was used to assess how the hypothesized organization of a set of identified factors fits the data. A bi-factor model was tested to check the possibility of a strong single factor (i.e., factor G), in addition to group factors that involve clusters of measures but are arguably separate from the general factor. All group factors were assumed to be uncorrelated to each other and variances of items could be explained by both group factors and the general factor in the bi-factor model. On the other hand, in the CFA, the correlations among the group factors were estimated and only group factors accounted for the variability of items. In our study, the

general factor, Factor G, was considered as SGMJ development because moral development is related to cognitive development in general [1,3,4,14,23–25]. According to Kohlberg, as cognitive capacities become more advanced, reasoning about moral situations becomes less self-focused, and therefore more developmentally advanced [1,3,4, 23].

Aligning with this perspective, Narvaez and Bock noted that individuals with more developmentally advanced moral reasoning possess intricate patterns of cognitive schemas, ultimately allowing for a greater perspective-taking ability when it comes to social decision-making [25]. Conversely, individuals with less developmentally advanced moral reasoning possess a much simpler pattern of cognitive schemas, resulting in a narrower perspective. Supporting this relationship is a wealth of data from DIT-2 studies that indicate a positive relationship between DIT-2 scores and cognitive capacities (reviewed as validity criteria by [2–4,14]). For example, Rest studied moral comprehension in relation to moral judgment development [26]. Moral comprehension refers to an individual's cognitive ability to comprehend moral concepts. Rest found evidence for a relationship between better moral comprehension skills and more developmentally advanced moral reasoning [26].

Moreover, based on Neo-Kohlbergian theory related to the soft-stage concept, we examined the presence of the general factor (the G factor or SGMJ), which is based on the assumption that moral judgment development occurs along a unidimensional continuum as proposed in the soft-stage model, by testing whether the bi-factor model fitted data better than the ordinary CFA model without the general factor. Unlike classical Kohlbergian theory that assumes qualitative transitions between different developmental levels (i.e. pre-conventional, conventional, and postconventional), Neo-Kohlbergian theory, which constitutes the basis of the DIT-2, argues that the development of moral judgment is continuous, quantitative, and similar to shifting distributions of judgment across different schemas [1,27]. Because the ordinary CFA model assumes three schemas as three independent latent factors, testing the ordinary model with CFA would not be an ideal measure to examine the soft-stage concept. Instead, the presence of the Factor G in the bi-factor model might be evidence that supports the soft-stage concept because the factor takes into account items across all three schemas and represents the SGMJ.

## Results

Descriptive statistics for the DIT-2 items and total score were presented in Table 1. The correlation matrix among the all DIT-2 items was provided as a S1 File because replication studies for CFA and bi-factor models are available using means, SDs, and correlation matrix for all items. The Cronbach's alpha was .840, indicating good reliability. The assumption of multivariate normality on the 51 items was checked using tests of multivariate skewness and kurtosis and the omnibus test of multivariate normality. The Small's test, Srivastava's test, and Mardia's test using DeCarlo's SPSS macro [28] indicated that multivariate normality was violated.

The unweighted least square (ULS) estimation method was selected because it is usually classified as a distribution-free-method and our data was not normally distributed [29]. The three-factor variances for CFA and general factor variance for the bi-factor model were fixed to 1.0 for scale identification [30]. This was appropriate because we usually take care of scale identification issues by assuming that factors are in z-score form, which would mean that their variances are 1.0 and this is what the LISREL does by default [31].

To find the acceptable model fit, four model fit indices were used: root mean square residual (RMSR), standardized root mean square residual (SRMSR), goodness-of-fit statistic (GFI), and the adjusted goodness-of-fit statistic (AGFI). The RMSR and SRMSR values less than 0.05 suggest good fit and values up to 0.08 are deemed acceptable [32,33]. Traditionally an adequate

**Table 1. Descriptive statistics for the DIT-2 (N = 44,742).**

| Item | n | Mean | SD |
|---|---|---|---|
| PI-1 | 44619 | 2.83 | 1.21 |
| PI-2 | 44666 | 3.82 | 1.11 |
| PI-3 | 44564 | 3.29 | 1.28 |
| PI-4 | 44638 | 2.82 | 1.27 |
| PI-5 | 44690 | 2.52 | 1.19 |
| PI-6 | 44631 | 2.45 | 1.13 |
| PI-7 | 44639 | 4.36 | 0.88 |
| PI-8 | 44493 | 2.30 | 1.09 |
| PI-9 | 44663 | 3.27 | 1.11 |
| PI-10 | 44664 | 2.23 | 1.09 |
| PI-11 | 44645 | 2.75 | 1.13 |
| PI-12 | 44692 | 3.56 | 1.09 |
| PI-13 | 44629 | 3.70 | 1.21 |
| PI-14 | 44597 | 3.23 | 1.34 |
| PI-15 | 44599 | 2.73 | 1.23 |
| PI-16 | 44597 | 2.96 | 1.25 |
| PI-17 | 44554 | 3.02 | 1.20 |
| PI-18 | 44523 | 3.31 | 1.29 |
| PI-19 | 44508 | 2.84 | 1.12 |
| PI-20 | 44440 | 2.79 | 1.24 |
| MN-1 | 44638 | 3.56 | 1.08 |
| MN-2 | 44640 | 3.54 | 1.27 |
| MN-3 | 44690 | 3.56 | 1.16 |
| MN-4 | 44450 | 2.84 | 1.07 |
| MN-5 | 44597 | 3.74 | 1.12 |
| MN-6 | 44537 | 3.31 | 1.23 |
| MN-7 | 44675 | 3.00 | 1.39 |
| MN-8 | 44615 | 3.23 | 1.16 |
| MN-9 | 44473 | 3.88 | 1.01 |
| MN-10 | 44451 | 3.54 | 1.13 |
| MN-11 | 44495 | 3.97 | 1.10 |
| MN-12 | 44595 | 3.54 | 1.47 |
| MN-13 | 44589 | 3.15 | 1.21 |
| MN-14 | 44505 | 3.52 | 1.40 |
| MN-15 | 44523 | 3.32 | 1.14 |
| MN-16 | 44475 | 3.41 | 1.10 |
| MN-17 | 44517 | 3.72 | 1.19 |
| P-1 | 44384 | 3.36 | 1.15 |
| P-2 | 44637 | 3.26 | 1.30 |
| P-3 | 44612 | 3.26 | 1.23 |
| P-4 | 44580 | 3.89 | 1.16 |
| P-5 | 44556 | 3.35 | 1.22 |
| P-6 | 44409 | 3.77 | 1.17 |
| P-7 | 44476 | 3.81 | 1.05 |
| P-8 | 44678 | 3.86 | 1.06 |
| P-9 | 44611 | 3.64 | 1.29 |
| P-10 | 44560 | 3.48 | 1.22 |

(*Continued*)

**Table 1.** (Continued)

| Item | n | Mean | SD |
|------|------|------|------|
| P-11 | 44627 | 4.05 | 1.09 |
| P-12 | 44431 | 3.66 | 1.20 |
| P-13 | 44476 | 3.54 | 1.20 |
| P-14 | 44448 | 3.25 | 1.29 |
| Total | 44742 | 165.86 | 19.94 |

Item scores range from 1 = No to 5 = great.

cut-off point of 0.90 is recommended for the GFI and AGFI [34]. A bi-factor model using the original item structure detailed in the DIT-2 Guide fitted our dataset better (see Table 2).

When an acceptable model fit was found, the next step was to determine significant parameter estimates. Unfortunately, we could not calculate the t value which is calculated by dividing the parameter estimate by the standard error. This was because ULS estimation did not produce standard errors [29]. Tables 3 and 4 describe the standardized factor loadings from the CFA and bi-factor model. Each item belonged to each schema well using a 3-factor CFA model and all factor loadings except those for four MN items (MN1, 2, 12, and 17), which were greater than 0.3 on the CFA. The hypothesis tests for factor loadings were not performed because the ULS estimation did not report the t-test results for parameters (see Table 3).

For the bi-factor model, most factor loadings of P items were higher on Factor G (SGMJ) than on Factor 3 (P). Ten MN items among 17 items had higher factor loadings on Factor 2 (MN) than Factor G (SGMJ), four MN items had lower factor loadings on Factor 2 than Factor G and three items had similar factor loadings between Factor G and MN. All factor loadings of PI items except three items (PI-12 to PI-14) on Factor 1 (PI) were much higher than Factor G's factor loadings.

We focused on items that reported standardized factor loadings on Factor G greater than 0.30, which has been regarded as a threshold for a meaningful factor loading [35,36]. The factor loadings of all P items exceeded this threshold (see Table 4 for the factor loading in the bi-factor model). In addition, two PI (PI-12 and PI-14) and three MN items (MN-5, MN-7, and MN-14) reported factor loadings greater than this threshold. As presented, interestingly, Factor G, SGMJ, was positively associated with PI and MN items although Kohlbergian and Neo-Kohlbergian theorists have assumed that the utilization of the postconventional schema indicates moral judgment development. In addition, in the bi-factor model, as shown in Table 4, all P items showed strong factor loadings on Factor G, while their factor loading on Factor 3 (P) became very small.

In addition, as a way to examine how each item is associated with the N2 score, we conducted correlation analysis. The resultant correlation coefficients are reported in Table 5. Consistent with the results from the bi-factor CFA, all P items were significantly associated with the N2 score, while several non-P items (i.e., PI-12, PI-14, MN-5, MN-7, MN-14) that reported significant factor loadings (see Table 4) also showed significant correlation.

**Table 2. Goodness-of-fit indices for DIT-2 (N = 39,409).**

| Model | RMSR | SRMSR | GFI | AGFI |
|-------|------|-------|------|------|
| CFA | 0.10 | 0.07 | 0.85 | 0.84 |
| Bi-factor model | 0.08 | 0.06 | 0.91 | 0.89 |

**Table 3. Standardized factor loadings from a 3-factor confirmatory factor analysis.**

| Item | Factor 1 (PI) | Factor 2 (MN) | Factor 3 (P) |
|---|---|---|---|
| PI-1 | 0.387 | | |
| PI-2 | 0.393 | | |
| PI-3 | 0.381 | | |
| PI-4 | 0.347 | | |
| PI-5 | 0.357 | | |
| PI-6 | 0.399 | | |
| PI-7 | 0.313 | | |
| PI-8 | 0.350 | | |
| PI-9 | 0.462 | | |
| PI-10 | 0.405 | | |
| PI-11 | 0.386 | | |
| PI-12 | 0.365 | | |
| PI-13 | 0.329 | | |
| PI-14 | 0.367 | | |
| PI-15 | 0.411 | | |
| PI-16 | 0.460 | | |
| PI-17 | 0.421 | | |
| PI-18 | 0.432 | | |
| PI-19 | 0.448 | | |
| PI-20 | 0.384 | | |
| MN-1 | | 0.221 | |
| MN-2 | | 0.278 | |
| MN-3 | | 0.371 | |
| MN-4 | | 0.429 | |
| MN-5 | | 0.492 | |
| MN-6 | | 0.452 | |
| MN-7 | | 0.356 | |
| MN-8 | | 0.400 | |
| MN-9 | | 0.380 | |
| MN-10 | | 0.368 | |
| MN-11 | | 0.411 | |
| MN-12 | | 0.250 | |
| MN-13 | | 0.386 | |
| MN-14 | | 0.392 | |
| MN-15 | | 0.313 | |
| MN-16 | | 0.316 | |
| MN-17 | | 0.294 | |
| P-1 | | | 0.357 |
| P-2 | | | 0.462 |
| P-3 | | | 0.411 |
| P-4 | | | 0.530 |
| P-5 | | | 0.462 |
| P-6 | | | 0.542 |
| P-7 | | | 0.450 |
| P-8 | | | 0.461 |
| P-9 | | | 0.515 |
| P-10 | | | 0.505 |

(*Continued*)

**Table 3.** (Continued)

| Item | Factor 1 (PI) | Factor 2 (MN) | Factor 3 (P) |
|---|---|---|---|
| P-11 | | | 0.552 |
| P-12 | | | 0.556 |
| P-13 | | | 0.458 |
| P-14 | | | 0.476 |

*Correlations among Factors*

| Factor | PI | MN | P |
|---|---|---|---|
| PI | 1.000 | | |
| MN | 0.364 | 1.000 | |
| P | 0.245 | 0.521 | 1.000 |

## Discussion

In general, we found that the bi-factor model reported a better model fit compared with the ordinary three-factor model in our CFA. These results suggest that the overall development of moral judgment can be better explained in terms of SGMJ that is represented by Factor G, rather than preferences on specific individual schemas that were represented by the three latent factors, PI, MN, and P. Also, the results support the soft-stage model, which constitutes the conceptual basis of the DIT-2, as proposed by Neo-Kohlbergians. Most factor loadings of P items were higher on Factor G (SGMJ) than on Factor 3 (P) in the bi-factor model. It supports the assumption that Factor G describes the unidimensional continuous moral judgment development. This might be because the postconventional schema, or the most developmentally advanced form of moral reasoning [1,3,4] is highly correlated with advanced cognitive development [1,3,4,23,24].

In addition, two PI and three MN items also showed significant factor loadings on Factor G in addition to all P items, so Factor G seemed to at least partially take into account PI and MN schemas in addition to the P schema. These five items showed significant positive correlation with the N2 score (see Table 5), which indicates the overall moral judgment development in Neo-Kohlbergian framework, unlike other PI and MN items. The N2 score incorporates all three schemas in its calculation and is based on the soft-stage model that assumes the gradual and incremental transitions across schemas instead of radical moves. Hence, the association between these five items and SGMJ may suggest that the bi-factor model and concept of SGMJ can show us how the soft-stage model is supported by evidence, although not all PI and MN items were associated with SGMJ. The nature of these five specific PI and MN items perhaps contributed to their significant association with SGMJ, unlike the other PI and MN items, so further details regarding these individual items will be discussed later in this section. Although the factor loadings of P items on the P factor diminished in the bi-factor model, the result does not necessarily suggest that the postconventional schema is not fundamental in explaining moral judgment development. Because all P items showed significant factor loadings on Factor G, and Factor G was fundamentally explained by P items, postconventional moral reasoning might constitute the basis of developed SGMJ and the explanation of moral judgment development in general. Given the better model fit indicators from the bi-factor model compared with the ordinary CFA model, and the strong factor loading of P items on Factor G, the soft-stage concept proposed in the Neo-Kohlbergian model can be better supported by evidence.

Moreover, in terms of the nature of the collected samples, these results could be interpreted based on the nature of the participants in the present study and prior research in moral development. We collected data from college students, who were situated within late adolescence to

**Table 4. Standardized factor loadings from a bi-factor model.**

| Item | Factor 1 (PI) | Factor 2 (MN) | Factor 3 (P) | Factor G (SGMJ) |
|------|---------------|---------------|--------------|-----------------|
| PI-1 | 0.516 | | | -0.072 |
| PI-2 | 0.389 | | | 0.111 |
| PI-3 | 0.287 | | | 0.229 |
| PI-4 | 0.296 | | | 0.175 |
| PI-5 | 0.471 | | | -0.088 |
| PI-6 | 0.501 | | | -0.057 |
| PI-7 | 0.269 | | | 0.126 |
| PI-8 | 0.401 | | | 0.011 |
| PI-9 | 0.453 | | | 0.107 |
| PI-10 | 0.550 | | | -0.102 |
| PI-11 | 0.317 | | | 0.166 |
| PI-12 | 0.140 | | | **0.418** |
| PI-13 | 0.161 | | | 0.280 |
| PI-14 | 0.198 | | | **0.353** |
| PI-15 | 0.375 | | | 0.180 |
| PI-16 | 0.525 | | | -0.015 |
| PI-17 | 0.454 | | | 0.011 |
| PI-18 | 0.424 | | | 0.116 |
| PI-19 | 0.455 | | | 0.062 |
| PI-20 | 0.316 | | | 0.210 |
| MN-1 | | 0.394 | | 0.046 |
| MN-2 | | 0.172 | | 0.197 |
| MN-3 | | 0.295 | | 0.195 |
| MN-4 | | 0.268 | | 0.255 |
| MN-5 | | 0.296 | | **0.322** |
| MN-6 | | 0.301 | | 0.259 |
| MN-7 | | 0.137 | | **0.308** |
| MN-8 | | 0.184 | | 0.285 |
| MN-9 | | 0.285 | | 0.213 |
| MN-10 | | 0.152 | | 0.283 |
| MN-11 | | 0.486 | | 0.163 |
| MN-12 | | 0.435 | | 0.014 |
| MN-13 | | 0.505 | | 0.125 |
| MN-14 | | 0.058 | | **0.463** |
| MN-15 | | 0.335 | | 0.058 |
| MN-16 | | 0.481 | | 0.052 |
| MN-17 | | 0.581 | | 0.000 |
| P-1 | | | 0.034 | **0.336** |
| P-2 | | | 0.494 | **0.398** |
| P-3 | | | 0.401 | **0.338** |
| P-4 | | | 0.089 | **0.521** |
| P-5 | | | 0.050 | **0.437** |
| P-6 | | | 0.038 | **0.570** |
| P-7 | | | 0.064 | **0.415** |
| P-8 | | | 0.087 | **0.448** |
| P-9 | | | 0.079 | **0.520** |
| P-10 | | | 0.074 | **0.499** |

(*Continued*)

**Table 4.** (Continued)

| Item | Factor 1 (PI) | Factor 2 (MN) | Factor 3 (P) | Factor G (SGMJ) |
|------|---------------|---------------|--------------|-----------------|
| P-11 |               |               | 0.092        | **0.532**       |
| P-12 |               |               | 0.251        | **0.509**       |
| P-13 |               |               | 0.172        | **0.422**       |
| P-14 |               |               | -0.003       | **0.489**       |

Bolded factor loadings on Factor G indicate factor loadings greater than the set threshold of .30.

emerging adulthood and likely to experience a transition from the conventional to postconventional moral thinking [4]. According to Kohlbergian theory, young college students are likely to temporally employ seemingly concrete individualistic and relativistic moral thinking that is similar to pre-conventional moral thinking before transitioning to the postconventional level [37,38]. Their responses suggest that they are attempting to integrate moral values into their own belief systems while dealing with internal conflicts and reflecting upon existing social norms to advance to the postconventional moral thinking. Such a trend is not necessarily evidence of the real retrogression of the development of moral judgment [37].

Thus, given that the majority of our participants were college students and likely to experience the aforementioned process explained by Kohlberg and Kramer [37], our participants might carefully consider and value several PI items and such PI items might show significant association with Factor G. Furthermore, because participants were experiencing a transition between the conventional to postconventional moral thinking [4], several MN items might

**Table 5. Spearman correlations between N2 score and individual items.**

| Item | $r_s$ | Item | $r_s$ | Item | $r_s$ |
|------|-------|------|-------|------|-------|
| PI-1  | -.359** | MN-1  | .034**  | **P-1**  | **.331**   |
| PI-2  | -.207** | MN-2  | .054**  | **P-2**  | **.404**   |
| PI-3  | -.083** | MN-3  | .031**  | **P-3**  | **.366**   |
| PI-4  | -.085** | MN-4  | .046**  | **P-4**  | **.469**   |
| PI-5  | -.333** | **MN-5**  | **.114**   | **P-5**  | **.325**   |
| PI-6  | -.327** | MN-6  | .056**  | **P-6**  | **.428**   |
| PI-7  | -.126** | **MN-7**  | **.097**   | **P-7**  | **.322**   |
| PI-8  | -.228** | MN-8  | .037**  | **P-8**  | **.371**   |
| PI-9  | -.212** | MN-9  | -.055** | **P-9**  | **.419**   |
| PI-10 | -.324** | MN-10 | -.012*  | **P-10** | **.366**   |
| PI-11 | -.111** | MN-11 | -.076** | **P-11** | **.432**   |
| **PI-12** | **.108**  | MN-12 | -.128** | **P-12** | **.449**   |
| PI-13 | -.084** | MN-13 | -.010*  | **P-13** | **.328**   |
| **PI-14** | **.034**  | **MN-14** | **.233**   | **P-14** | **.369**   |
| PI-15 | -.052** | MN-15 | -.232** |          |            |
| PI-16 | -.318** | MN-16 | -.120** |          |            |
| PI-17 | -.328** | MN-17 | -.138** |          |            |
| PI-18 | -.158** |       |         |          |            |
| PI-19 | -.233** |       |         |          |            |
| PI-20 | -.028** |       |         |          |            |

* $p < .05$

** $p < .01$

also be seriously considered and show significant factor loadings on Factor G. As mentioned earlier, our factor analysis was performed at the item level and took into account all individual items, so it follows that our results will differ from those in the calculation of N2 scores, which utilize only four rank-ordered items per story. Because Factor G, SGMJ, was examined based on all individual items, the calculated factor loadings on SGMJ suggest that even participants with sophisticated moral judgment might also show partial preference for several non-P items, two PI and three MN items, in addition to P items.

We may also tentatively consider a possible association between Factor G and practical wisdom, *phronesis*, proposed in the Aristotelian theoretical framework. *Phronesis* enables one to choose the most appropriate solution in a dilemmatic situation [39,40]. It is an intellectual virtue to seek an optimal solution, so it is plausible to expect a positive association between *phronesis* and the P schema. However, according to Aristotelian virtue ethics, *phronesis* requires balancing diverse values and viewpoints that might also embrace communal, relational, and interdependent aspects that might not be fully captured by the P schema [41–43]. Thus, several PI and MN items, which seemingly were associated with the aforementioned communal, relational, and interdependent values, show significant factor loadings ($> .3$) on Factor G in the dataset. However, because we only employed the DIT-2 based on the Neo-Kohlbergian framework and did not directly test the association between moral judgment and wisdom, our interpretation from the Aristotelian perspective is tentative.

Furthermore, we found several interesting points in the reported factor loadings. As reported in Table 4, we found that several non-P items (i.e., PI-12. PI-14. MN-5, MN-7, MN-14) showed significant associations with the SGMJ, the Factor G. To consider the implications of these findings, we reviewed the nature of the aforementioned two PI (PI-12 and PI-14) and three MN items (MN-5, MN-7, and MN-14). The two PI items were concerned mainly with communal values (PI-12) or relational values (PI-14). In the cases of the three MN items, MN-5 seemed to be associated with the harmony in a community. MN-7 was seemingly concerned about the maintenance of communal order (e.g., laws), and MN-14 was related to one's rights in the relation with a community.

Compared with the other PI and MN items that were more about one's self-interest and social norms and conventions, respectively, these five items were more related to the welfare of other human beings and their rights in society. Hence, we could consider that these items address basic moral concerns in human and social relations, not one's own interests or social norms per se that are represented in other PI and MN items, and are likely to show the higher factor loadings and higher correlation with the N2 score. These items are perhaps more strongly associated with SGMJ because they would better examine the gradual transition from person- (PI) or social-oriented morality (MN) to principled morality (P) while the other PI and MN items might be less related to morality. In addition, the phrasing of the five items might also contribute to their significant association with the N2 score. These items seemed to imply moral philosophical ideas that were slightly more advanced than their respective levels. For instance, the phrasing of two PI items was about responsibility and empathic concern and three MN items were phrased in a way that required participants to consider the actors' rights and responsibilities instead of explicit laws or norms. Because phrases used in these items addressed ones' rights and obligations, which might be associated more with moral principles represented in the P schema than with personal values (PI) or laws or norms (MN), they perhaps showed a significant association with the N2 score. These anomalies may also be the result of the age/developmental level of the sample—it is more restricted than the full range (adolescent-adult) and perhaps these items were interpreted in a different way—that is, they read into the items.

Interestingly, when we examined how each item's rating was associated with the overall postconventional moral judgment in terms of N2 scores, we found that ratings on the aforementioned PI and MN items were positively correlated with N2 scores while all P items were positively correlated with N2 scores as expected (see Table 5). When we examined the correlation, PI-12 and PI-14 showed the highest positive correlation coefficients with N2 scores among all PI items, and MN-5, MN-7, and MN-14 did so among all MN items (see Table 5). Because the N2 score is calculated with only four rank-ordered items per story [2], it would be difficult to examine whether and how items other than these four items per story are assessed and valued by participants. In fact, although participants can score more than five items as "5 = great" in a certain story, only four items would be included in calculating their N2 scores. For instance, a hypothetical participant who selected all P items for the rank-order questions might receive an ideal N2 score, but the participant could also mark some PI and MN items as "5 = great". The CFA with all individual items that we performed would enable us to examine the relationship between moral judgment in general, SGMJ in our study, and the individual items that could not be understood with the N2 scores. Given that the five items showed the highest correlation coefficients with N2 scores within their schema categories, PI and MN, they were likely to show significant factor loadings on Factor G. Moreover, these five PI and MN items might be the most preferred PI and MN items among participants with relatively more developed judgment.

However, some limitations warrant future research. First, the present factor analyses are based on undergraduate college students only, but it could prove fruitful to investigate the internal factor structure of the DIT-2 for different populations, such as high school students and graduate students. Second, only rating items were used for this validity study, although DIT-2 reports N2 scores calculated from both ratings and rankings [14]. Third, there are some low and weak correlations between items, which is possibly due to college students being in transition in their use of moral judgment developmental schemas. When a heterogeneous population including middle, high school, and graduate level students is analyzed, the correlation might increase.

In conclusion, a bi-factor model using the original item structure in the manual would be proposed for undergraduate populations. A simpler model, CFA, was also considered to evaluate the internal structure of DIT-2 scale. The greater than 0.3 factor loadings for all 51 items, except four MN items, indicated that our internal structure from the DIT-2 manual worked well using the CFA. For the bi-factor model, the most P items were loaded more on Factor G (SGMJ) than on Factor 3 (P). We concluded that a DIT-2 scale is a valid measure of the internal structure of moral reasoning development from both CFA and bi-factor models. In addition, we attempted to interpret the meaning of the finding, the presence of Factor G and its significant association with several PI and MN items, with prior research on moral judgment development.

## Supporting information

**S1 File.**
(XLSX)

## Author Contributions

**Conceptualization:** Youn-Jeng Choi, Hyemin Han.

**Data curation:** Stephen J. Thoma.

**Formal analysis:** Youn-Jeng Choi, Hyemin Han.

**Investigation:** Youn-Jeng Choi, Stephen J. Thoma.

**Methodology:** Youn-Jeng Choi, Hyemin Han.

**Project administration:** Youn-Jeng Choi.

**Supervision:** Youn-Jeng Choi.

**Validation:** Youn-Jeng Choi, Hyemin Han.

**Visualization:** Youn-Jeng Choi.

**Writing – original draft:** Youn-Jeng Choi, Hyemin Han, Meghan Bankhead, Stephen J. Thoma.

**Writing – review & editing:** Youn-Jeng Choi, Hyemin Han, Meghan Bankhead, Stephen J. Thoma.

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
