## [Decision Letter · Decision Letter 0]

22 Apr 2020

PONE-D-19-32500

Validity Study using Factor Analyses on Defining Issues Test-2 in Undergraduate Populations

PLOS ONE

Dear Dr. Choi,

Thank you for submitting your manuscript to PLOS ONE. After careful consideration, we feel that it has merit but does not fully meet PLOS ONE’s publication criteria as it currently stands. Therefore, we invite you to submit a revised version of the manuscript that addresses the points raised during the review process.

We would appreciate receiving your revised manuscript by May 21, 2020. To enhance the reproducibility of your results, we recommend that if applicable you deposit your laboratory protocols in protocols.io, where a protocol can be assigned its own identifier (DOI) such that it can be cited independently in the future. For instructions see: http://journals.plos.org/plosone/s/submission-guidelines#loc-laboratory-protocols

We look forward to receiving your revised manuscript.

Kind regards,

Stephan Doering, M.D.

Academic Editor

PLOS ONE

Journal Requirements:

2. Thank you for including your ethics statement: "The DIT-2 datasets were collected by the Center for the Study of Ethical Development in the University of Alabama and the IRB was approved."

a) Please amend your current ethics statement to confirm that your named institutional review board or ethics committee specifically approved this study.

Reviewers' comments:

Reviewer's Responses to Questions

**Comments to the Author**

1. Is the manuscript technically sound, and do the data support the conclusions?

Reviewer #1: Partly

Reviewer #2: Partly

2. Has the statistical analysis been performed appropriately and rigorously? 

Reviewer #1: Yes

Reviewer #2: Yes

3. Have the authors made all data underlying the findings in their manuscript fully available?

Reviewer #1: No

Reviewer #2: Yes

4. Is the manuscript presented in an intelligible fashion and written in standard English?

Reviewer #1: Yes

Reviewer #2: Yes

5. Review Comments to the Author

Reviewer #1: The manuscript presents a psychometric validation study of DIT-2, a measure of moral judgement based on Neo-Kohlbergian cognitive-developmental theory by Rest et. al.

The study was based on a very large sample of undergraduate students from the USA. Methodologically and technically the study seems well conducted, but the presentation is poor. The current form of the manuscript is confusing and there are issues that should be clarified/corrected:

- The manuscript does not follow the standard IMRaD structure (there is no Discussion section at all) which makes it hard to follow. Some issues addressed in the Results section should actually be a Discussion. There is no broader discussion of the study results.

- The purpose of the study is not presented in a clear manner. It is not clear what is meant by "internal structure" of the DIT-2 and how it is different from the validity of the measure. Later, at the end of the Methods section authors state that "the presence of the G-factor in the bi-factor model might be evidence that supports the soft stage concept as the factor takes into account items accross all three schemes and represents SGMJ". This statement sounds like the aim of the study was also to test the underlying theory.

- The underlying logic of the study is not clear. After reading the manuscript, I cannot tell what this study means. How are the obtained results relevant (do they add to the theory? do they offer an option to improve the measure?) and how does the G-factor relate to the three schema-factors. Maybe graphic representation of the tested models would add some clarity (?).

- The scheme-general moral judgement (SGMJ) factor is poorly explained and, looking at the results, it is difficult to understand how they support the soft-stage hypothesis. If there was a G-factor, shouldn't most of the items (not just P and very few of MN and PI) show meaningful factor loadings? I don't think that the manuscript addresses this issue adequately.

- Although it is true that the bi-factor model performed somewhat better, the two tested models were very similar. This is not addressed in the manuscript.

- p. 12 "...factor G seemed to take into account all schemes instead of solely focusing on postconventional scheme." This is a strong claim given that all P items had saturation >.3 on G and only 2 out of 20 PI items and 3 out of 17 MN items.

- What is the meaning of the finding that only PI and MN items addressing to communal and relational values loaded on G?

Reviewer #2: General comments

The take-home message of this manuscript should be more clearly stated. To this reviewer the main message seems to be that bi-factor modelling revealed the presence of factor G, which provides evidence for the Neo-Kohlbergian soft-stage concept. If this is correct, then it is an interesting finding. However, the reviewer feels that the manuscript in its current form is not suitable for publication. Firstly, it fails to provide sufficient background that would contextualize it and allows readers outside the field to understand the purpose and significance of the study. Secondly, it does not strictly follow the structure of a primary research article (e.g. what belongs in the Introduction is stated under Methods, Introduction and Conclusion are too brief, etc – please see comments for details). In addition, given the highly specialized nature of this study, this reviewer feels that it would be better suited for a more specialized journal in the field of moral development.

Abstract:

1. At 127 words, the Abstract is too brief (127 words!). It does not convince the reader that the information being communicated is important. Some questions that the authors might consider answering are in this section are: Why was there a need to do this study? How was the study carried out? What key aspects/limitations of the study would the reader need to be aware of? What exactly was done? What were the findings? How does the knowledge that has been generated serve to advance the field?

2. “Confirmatory factor analysis” is mentioned twice in the first and second sentences. Consider revising these sentences to avoid repetition.

3. In the sentence “A 3-factor confirmatory factor analysis based on the DIT-2 manual and bi-factor models were evaluated for model fit”, is it a bi-factor model or models? It should be the former, right?

4. How does this study contribute to research on moral development? For example, why was there a need to undertake this test of internal validity? To assess dimensionality or?

5. Take-home message should be stated clearly. For example, was the study undertaken to show that the strong factor loading of P items on Factor G provides evidence to support the soft-stage concept within the Neo-Kohlbergian model?

Introduction:

1. As with the Abstract, the Introduction is not comprehensive. There is not enough background to provide context that would allow readers outside the field to understand the purpose and significance of the study. The problem statement is weak. Numerous references are cited but not much elaboration is provided, rendering the literature review as a whole inadequate. The aim and significance of the study are not stated clearly. Why was there a need to test the internal validity of the DIT-2? A debate, controversy, or unresolved problem in the field?

2. The DIT definition is too brief (“The Defining Issues Test (DIT) is a test concerned with how one defines the moral issues in a social problem (Rest, Narvaez, Bebeau, & Thoma 1999; Rest, Narvaez, Thoma, & Bebeau, 1999a; Rest, Narvaez, Thoma, & Bebeau 1999b; Rest, Narvaez, Thoma, & Bebeau, 2000)). Since the authors cite many references, can they provide a more informative definition?

3. What makes the DIT so popular?

4. What is the difference between DIT-1 and DIT-2? Why have you chosen to focus on DIT-2?

5. “Much less is known about the validity as it relates to the internal structure of the DIT”. Why is this a problem exactly? Can the authors state the purpose of this study more clearly? Was it done to corroborate previous validity studies (if any)? Or is it expected to lead to a new rubric for measuring moral development?

6. “The purpose of this study is to validate the internal structure of DIT-2….The purpose is apparently stated again under Methods, p. 9: “Moreover, based on Neo-Kohlbergian theory related to the soft stage concept, we intended to examine the presence of the general factor, the G factor or SGMJ, by testing whether the bi-factor model was better than the ordinary CFA model without the general factor.” (see comment on this same sentence under Methods below).

7. Provide a general description of 1) CFA 2) bi-factor fitting model. What are the advantages of CFA and the bi-factor model?

8. What is the relationship between internal structure and interpretation/proposed uses of test scores? Why is internal structure important?

9. 3 sentences begin with “One” (i.e. One who prefers the personal interests scheme…One who prefers the maintaining norms scheme…One who prefers the postconventional scheme…). Please re-phrase.

10. “Researchers conducting the previous validity studies (Davison, 1979; Davison, Robbins, & Swanson, 1978; Ma & Chan, 2001) used exploratory factor analysis (EFA) with the principal component extraction method for the DIT-1, though the DIT-2 was not explored in this manner.”

a. This sentence should begin a new paragraph.

b. Which theoretical model did these prior studies consider? Was it different from the current schema model?

Method

1. The authors provide one example question: “Isn’t the doctor obligated by the same laws as everybody else if giving an overdose would be the same as killing her?” Can the authors enclose the corresponding dilemma in an appendix, along with examples of the ranking questions?

2. Can the authors enclose the questionnaire?

3. Was any assumption check done for factor analysis?

4. p.7: “To prevent culture and language bias, only the students who are (change to “were”) US citizens and use (change to “used”) English as their primary language were selected.”

5. Ages range (change to “ranged”) from 17 to 26 and 47.2% of participants are (change to “were”) male (n = 21,139) and 52% are (change to “were”) female (n =23,272).

6. p.8: Each item was assumed to measure one of three specific schemas of moral reasoning (i.e., PI, MN and P schemas). By “item” do you mean “survey item”?

7. p.8: The authors should introduce the DIT-2 Guide in the Introduction.

8. Tabulate the 51 rating item assignments.

9. Specify that PI, MN, and P are the specific factors and Factor G is the general factor.

10. p.8: The second paragraph (“A bi-factor model was tested to check…) and third paragraph (Aligning with this perspective,…”) belong in the Introduction.

11. What is the definition of scheme general moral judgement (SGMJ)? Did you coin this term yourselves or you citing somebody else? If the latter, please provide the reference(s).

12. p.9: “soft stage concept”. What does this mean and what background references can you provide? Please include this in the Introduction.

13. p.9 Shouldn’t this statement of motivation be in the Introduction instead? “Moreover, based on Neo-Kohlbergian theory related to the soft stage concept, we intended to examine the presence of the general factor, the G factor or SGMJ, by testing whether the bi-factor model was better than the ordinary CFA model without the general factor.”

Results

1. Did the authors test competing CFA models? If so, can they summarize the comparison here?

2. p.13: “…aforementioned (which ones???) PI and MN items were positively correlated with N2 scores while all P items were positively correlated with N2 scores as expected (see Table 5).”

3. p.11-12: The paragraph on the 2 PI and 3 MN items whose factor loadings onto Factor G were greater than the threshold ends abruptly on p.12. Can the authors either indicate that they discuss the implications later in the section or re-organize the section?

4. p.12: Sudden mention of N2 scores here. Unless the N2 score is first described in the Introduction and/or Methods, the reader will not be able to appreciate the rationale for using the N2 score to determine the correlation between PI-12, PI-14, MN-5, MN-7, and MN-14 with “more developed moral judgement.”

5. Did the authors consider whether the phrasing of the PI-12, PI-14, MN-5, MN-7, and MN-14 items might explain their positive correlations with N2 scores?

Conclusion

1. What are the limitations of the bi-factor model?

2. What are the (theoretical) implications of this work for the construct (moral judgement)?

3. How does this study advance the field? Will this study help to provide a new and modified rubric to measure moral judgment development? Will it strengthen the results of previous validity studies? Have similar analyses been done before?

References

p. 19: The Ma, H. K. & Chan, W. S. reference was published in 1987, not 2001!

6. PLOS authors have the option to publish the peer review history of their article (what does this mean?). If published, this will include your full peer review and any attached files.

Reviewer #1: Yes: Darko Hren

Reviewer #2: No

---

## [Author Response · Author response to Decision Letter 0]

26 Jun 2020

Dear the PLOS One editor and reviewers,

We sincerely appreciate your invaluable comments and suggestions on our manuscript. We have tried to be responsive to your comments and have incorporated our changes in the revised version. I hope you find significant improvements from the revised manuscript.

Best regards,

Youn-Jeng Choi & Hyemin Han, PhD

---

## [Decision Letter · Decision Letter 1]

27 Jul 2020

PONE-D-19-32500R1

Validity study using factor analyses on the Defining Issues Test-2 in undergraduate populations

PLOS ONE

Dear Dr. Choi,

Thank you for submitting your manuscript to PLOS ONE. After careful consideration, we feel that it has merit but does not fully meet PLOS ONE’s publication criteria as it currently stands. Therefore, we invite you to submit a revised version of the manuscript that addresses the points raised during the review process.

We look forward to receiving your revised manuscript.

Kind regards,

Stephan Doering, M.D.

Academic Editor

PLOS ONE

Reviewers' comments:

Reviewer's Responses to Questions

**Comments to the Author**

1. If the authors have adequately addressed your comments raised in a previous round of review and you feel that this manuscript is now acceptable for publication, you may indicate that here to bypass the “Comments to the Author” section, enter your conflict of interest statement in the “Confidential to Editor” section, and submit your "Accept" recommendation.

Reviewer #1: All comments have been addressed

Reviewer #2: All comments have been addressed

2. Is the manuscript technically sound, and do the data support the conclusions?

Reviewer #1: Partly

Reviewer #2: Yes

3. Has the statistical analysis been performed appropriately and rigorously? 

Reviewer #1: I Don't Know

Reviewer #2: Yes

4. Have the authors made all data underlying the findings in their manuscript fully available?

Reviewer #1: No

Reviewer #2: Yes

5. Is the manuscript presented in an intelligible fashion and written in standard English?

Reviewer #1: Yes

Reviewer #2: Yes

6. Review Comments to the Author

Reviewer #1: The authors have answered all my comments.

I am still not fully convinced by the interpretation of the results. It seems to me that the G-factor they got in the "bi-factor" model is some form of "higher level" P schema. I may be wrong. But if I am, the explanation of the results needs clarification for non-expert readers.

- The language and style of the manuscript would benefit from some quality editing.

- The term "bi-factor model" is confusing because it can imply two factors and not two levels of factors. Maybe "two-level model" or "bi-level model" would be a better term (?) Just a suggestion. Disregard if it makes no sense.

- The last sentence of the Introduction is unnecessary. It is understood that implications of the results will be discussed.

Reviewer #2: I thank the authors for addressing all of my comments. The manuscript has improved significantly since the first round. However, I do have some additional concerns that they will need to address before this manuscript can be published - I refer the authors to my comments and corrections in the attachment. Generally, the authors need to be more careful about typos, tenses, long-winded sentences, and in-text citations. They should also avoid cumbersome phrases.

7. PLOS authors have the option to publish the peer review history of their article (what does this mean?). If published, this will include your full peer review and any attached files.

Reviewer #1: **Yes: **Darko Hren, PhD, Assoc Prof

Reviewer #2: No

---

## [Author Response · Author response to Decision Letter 1]

6 Aug 2020

Dear the PLOS One reviewers,

We sincerely appreciate your invaluable comments and suggestions on our manuscript. We have tried to be responsive to your comments and have incorporated our changes in the revised version. We could not improve our manuscript without your careful and significant feedback.

Best regards,

Youn-Jeng Choi & Hyemin Han, PhD

---

## [Editor Report · Decision Letter 2]

11 Aug 2020

Validity study using factor analyses on the Defining Issues Test-2 in undergraduate populations

PONE-D-19-32500R2

Dear Dr. Choi,

We’re pleased to inform you that your manuscript has been judged scientifically suitable for publication and will be formally accepted for publication once it meets all outstanding technical requirements.

Kind regards,

Stephan Doering, M.D.

Academic Editor

PLOS ONE

---

## [Editor Report · Acceptance letter]

17 Aug 2020

PONE-D-19-32500R2 

Validity study using factor analyses on the Defining Issues Test-2 in undergraduate populations 

Dear Dr. Choi:

I'm pleased to inform you that your manuscript has been deemed suitable for publication in PLOS ONE. Congratulations! Your manuscript is now with our production department. 

Kind regards, 

on behalf of

Professor Stephan Doering 

Academic Editor

PLOS ONE